# Relative Uncertainty Learning for Facial Expression Recognition

**Yuhang Zhang, Chengrui Wang, Weihong Deng**
Beijing University of Posts and Telecommunications
zyhzyh@bupt.edu.cn, crwang@bupt.edu.cn, whdeng@bupt.edu.cn

## Abstract

In facial expression recognition (FER), the uncertainties introduced by inherent noises like ambiguous facial expressions and inconsistent labels raise concerns about the credibility of recognition results. To quantify these uncertainties and achieve good performance under noisy data, we regard uncertainty as a relative concept and propose an innovative uncertainty learning method called Relative Uncertainty Learning (RUL). Rather than assuming Gaussian uncertainty distributions for all datasets, RUL builds an extra branch to learn uncertainty from the relative difficulty of samples by feature mixup. Specifically, we use uncertainties as weights to mix facial features and design an add-up loss to encourage uncertainty learning. It is easy to implement and adds little or no extra computation overhead. Extensive experiments show that RUL outperforms state-of-the-art FER uncertainty learning methods in both real-world and synthetic noisy FER datasets. Besides, RUL also works well on other datasets such as CIFAR and Tiny ImageNet. The code is available at https://github.com/zyh-uaiaaaa/Relative-Uncertainty-Learning.

## 1  Introduction

Although well-trained models can give high-confidence inferences directly, the inference results may actually be wrong and cause serious consequences [22]. Therefore, the concept of uncertainty is widely used in measuring how well the model trusts its inference results, which has become a research hotspot in trustworthy machine learning [3, 11, 10, 22, 26, 16]. It is expected that a reliable model assigns high level uncertainty to its erroneous predictions so that humans can intervene to avoid lots of disasters. As described by Kiureghian *et al.* [25], the uncertainties in machine learning can be split into aleatoric uncertainty and epistemic uncertainty, which are also called data uncertainty and model uncertainty. In this paper, we focus on quantifying data uncertainty as model uncertainty can be eliminated by introducing more training data. Concretely, data uncertainty in facial expression recognition (FER) is mainly caused by ambiguous facial expressions and the subjectiveness of annotators, which blocks the improvement of recognition performance [47].

Recent studies propose a variety of methods to solve the uncertainty learning problem in the face recognition field. PFE [37] introduces a new branch to learn the uncertainty of face recognition and recognizes faces by measuring the similarity between two Gaussian distributions. Chang *et al.* [4] further propose DUL to learn feature and uncertainty simultaneously in order to improve facial feature learning. When it comes to FER, SCN [47] uses a fully-connected layer to learn an importance weight for each image and suppresses uncertainties according to the learned weights. However, the strong learning ability of neural networks will deteriorate the uncertainty learning branch. One solution is adding regularization to force the model to learn uncertainty, which needs to be tuned carefully, otherwise, it would lead to underfitting. For example, SCN uses margin loss to keep the gap between certain and uncertain images, but choosing a suitable gap value is non-trivial. DUL assumes features

to follow a Gaussian distribution, but the real distribution of the dataset is not always following such an assumption. Therefore, it is still a challenge to learn uncertainty distributions of real datasets without deteriorating feature learning.

To address the problem, we perceive data uncertainty from a different point of view. We realize that data uncertainty – the difficulty to classify a sample correctly is actually a relative concept. More specifically, the difficulty of classification is a person's subjective feeling that comes from comparison. One can only know whether a sample is easy or not according to a reference. Inspired by this idea, we propose an innovative method called Relative Uncertainty Learning (RUL) to help deep learning models to learn uncertainty for each sample. Concretely, we build a new branch to model the uncertainties of input images and utilize the uncertainties as weights to mix two features of different labels. Through an add-up loss function, the model is encouraged to recognize two expressions simultaneously from the mixed features, which enables the model to learn uncertainty comparatively while minimizing the total loss. As discussed in Section 3.3, the uncertainty learning branch will assign large uncertainty values to uncertain facial expression images while small uncertainty values to certain images.

The main contributions of this work are as follows:

- We perceive uncertainty from a new perspective and propose an innovative uncertainty learning method to learn uncertainty from the relative difficulty of two samples.

- We get state-of-the-art performance in both real-world and synthetic noisy FER datasets.

- RUL does not need prior knowledge of the dataset uncertainty distribution and can be easily applied to different classification tasks with low computation cost.

## 2   Related Work

**Uncertainty learning in the face field** Uncertainties can be mainly categorized into model uncertainty which accounts for uncertainty in the model parameters and data uncertainty which is the noise inherent in the dataset [22]. In the face recognition field, uncertainty learning is important for learning discriminative feature embeddings of noisy face images. Because noisy face images are usually out of the cluster and have large variances in the latent embedding space, which might cause wrong recognition [37]. There have been several proposals to solve the uncertainty learning problem in face recognition tasks [13, 51, 23, 37, 4]. Some of them [13, 51, 23] have tried to analyze and learn face representations using model uncertainty. Gong *et al.* [13] estimate the capacity of a given face representation by explicitly accounting for the manifold structure, model uncertainty and data variability. Khan *et al.* [23] present a framework for class imbalance learning based on model uncertainty. Zafar *et al.* [51] cope with false positives through employing model uncertainty to improve the efficacy of face recognition systems. Thereinto the methods considering data uncertainty learning. Shi *et al.* [37] use PFE to learn a variance for each fixed feature and then measure the likelihood of each positive face pair of $(x_i, x_j)$ sharing the same distribution of latent embedding p($z_i$, $z_j$). Chang *et al.* [4] further propose to learn feature and uncertainty simultaneously by encoding the latent feature embedding as a Gaussian distribution. In the facial expression recognition task, Wang *et al.* [47] try to learn an attention weight for each image, then utilize it to weigh cross-entropy loss in order to suppress the influence of noisy samples. She *et al.* [36] propose to use several branches to model the latent label distribution of facial expression images and use cosine similarity to capture the uncertain images.

**Noisy dataset training** With the enhancement of deep learning network capabilities, the datasets for facial expression recognition (FER) are getting larger and larger. For the large-scale FER datasets [43, 54, 7, 1, 33, 9, 27], due to the ambiguity of expressions and the subjectiveness of human annotators, it is extremely hard to get high-quality labels. Training with label noise has been studied for a long time. The basic idea is to enable the model to access the quality of the labels or estimate the noise distribution using a small set of clean dataset [5, 28, 39, 45]. Li *et al.* [28] design a distillation framework to use information from both a small clean dataset and label relations in knowledge graphs to prevent the model learning from noisy labels. Veit *et al.* [45] use a small amount of clean annotations to reduce the noisy samples before fine-tuning the network using both the clean set and the full set. Other methods do not use a small clean dataset, but usually assume extra constraints on

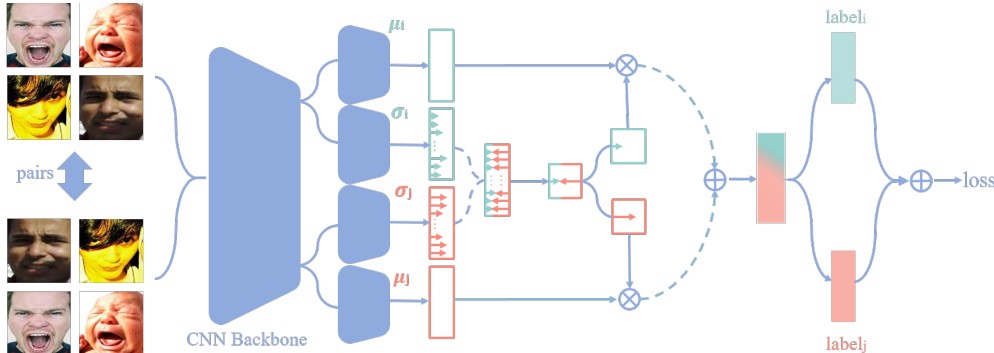

Figure 1: Overview of our Relative Uncertainty Learning method. We use two branches to learn facial features $\mu$ and uncertainty vectors $\sigma$ simultaneously. Then we normalize uncertainty vectors with their counterparts sampled from shuffled indexes. We compute the mean of the uncertainty vectors as the uncertainty values. Next, we mix the facial features according to the mean uncertainty values. *Note that the easy sample will get a small uncertainty value because its facial feature is very obvious for recognition. In contrast, the ambiguous sample tends to get a large uncertainty value in order to make it easy for the model to recognize its corresponding label from the mixed features.* Finally, we use the add-up loss to encourage the model to recognize the two expressions from the mixed feature, which implicitly forces the model to learn reliable uncertainty values.

the noisy samples. Natarajan *et al.* [34] design a specific loss function for randomly flipped labels. Goldberger *et al.* [12] explicitly model the noises in datasets by a softmax layer which connects the correct labels to the noisy ones. Without collecting a clean set of dataset or using prior knowledge of noise distribution, our method learns uncertainty through the relativity of the given dataset and suppresses the most uncertain samples to get a good performance under label noise.

## 3 Proposed Method

In order to reduce the negative impact of the strong learning ability of neural networks and learn uncertainty values closer to the real dataset uncertainty distribution, we propose a simple and effective method named Relative Uncertainty Learning (RUL). In this section, we first provide the motivation behind our method. Then we demonstrate the details of RUL. Finally, we explain why RUL is effective in uncertainty learning.

### 3.1 Motivation

The uncertainty learning methods mentioned before suffer from the strong learning ability of deep neural networks more or less. For example, the introduced variances in DUL are equivalent to some noises being added to feature points, so the model will learn very small variances for all feature points making the estimation of uncertainty less reliable. SCN also learns large importance weights for noisy samples to obtain small cross-entropy loss values as these noisy images will be memorized by the networks after training for several epochs.

In order to prevent the strong learning ability of neural networks from degrading the uncertainty learning branch, we perceive uncertainty from a relative point of view. We find that learning uncertainty through comparison forms a natural regularization because if the model assigns a very small uncertainty value for one image of the comparison pair, then it means the counterpart image gets a large uncertainty value which prevents the model learning small uncertainty values for all images. Learning uncertainty through relativity also conforms to human cognition – it is very difficult for a person to define which sample is certain and which sample is uncertain without using one image as a reference. In the following section, we thoroughly illustrate the details of our proposed RUL.

## 3.2 Overview of Relative Uncertainty Learning

In this section, we illustrate the structure of our proposed RUL. The pipeline of RUL is shown in Figure 1.

**Two Branches Model** Given a batch of face images $\boldsymbol{X}$, assuming the conventional feature extraction module $f_\theta(\boldsymbol{X})$ can be decomposed as

$$f_\theta(\boldsymbol{X}) = f_\theta^{(l)}(f_\theta^{(l-1)}(\boldsymbol{X})), \tag{1}$$

we first extract general facial features $\boldsymbol{F}_g$ from the penultimate feature extraction layer denoted as $\boldsymbol{F}_g = f_\theta^{(l-1)}(\boldsymbol{X})$ [50]. Then, in order to learn facial features and uncertainty values simultaneously, we drop the layer indexed by $l$, i.e., $f_\theta^{(l)}$ and build two separate branches $f_{\theta_\mu}(f_\theta^{(l-1)}(\boldsymbol{X}))$ and $f_{\theta_\sigma}(f_\theta^{(l-1)}(\boldsymbol{X}))$ to produce facial features $\boldsymbol{F} = [\boldsymbol{\mu}_1, \boldsymbol{\mu}_2, ..., \boldsymbol{\mu}_N] \in \mathbb{R}^{D \times N}$ and uncertainty vectors $\boldsymbol{U} = [\boldsymbol{\sigma}_1, \boldsymbol{\sigma}_2, ..., \boldsymbol{\sigma}_N] \in \mathbb{R}^{D \times N}$. $D, N$ denote output dimension and batch size separately.

**Relative Uncertainty Learning** Inspired by the relativity of the uncertainty concept and the mixup method [53, 42, 46], we mix two different facial features according to their uncertainty values which enables the FER model to learn uncertainty through the relativity of different samples. First, we mix a mini-batch with their shuffled ones which results in the facial images of the whole dataset being compared with each other during the training process because of the randomness introduced by the dataloader and the shuffle of mini-batches. Before mixing features, we first normalize the uncertainty vectors element-wise to compare two features with each other following Equation (2)

$$\hat{\boldsymbol{\sigma}}_i, \hat{\boldsymbol{\sigma}}_j = \frac{\boldsymbol{\sigma}_i}{\boldsymbol{\sigma}_i + \boldsymbol{\sigma}_j}, \frac{\boldsymbol{\sigma}_j}{\boldsymbol{\sigma}_i + \boldsymbol{\sigma}_j}, \tag{2}$$

$\boldsymbol{\sigma}_i$ and $\boldsymbol{\sigma}_j$ are the uncertainty vectors for image $i$ and its counterpart image $j$ sampled from the shuffled indexes, note that they have different labels. This operation not only forms a natural regularization as the model can not learn small uncertainty values for both features, but also benefits the training process as extreme uncertainty values might make the training process unstable or even the loss can not converge. As mentioned by DUL [4], the mean of the predicted uncertainty vectors $\hat{\boldsymbol{\sigma}}_i \in \mathbb{R}^D$ can be viewed as an approximated measurement of the estimated uncertainty for image $i$, we also use the mean of $\hat{\boldsymbol{\sigma}}_i$ as learned uncertainty value for image $i$ denoted as $\hat{\sigma}_{i_{mean}} \in \mathbb{R}$. Then we get an uncertainty value for each input image denoted as $\boldsymbol{U} = [\hat{\sigma}_{1_{mean}}, \hat{\sigma}_{2_{mean}}, ..., \hat{\sigma}_{N_{mean}}] \in \mathbb{R}^{1 \times N}$. We then mix features according to their uncertainty values following Equation (3)

$$\tilde{\boldsymbol{\mu}} = \hat{\sigma}_{i_{mean}} \boldsymbol{\mu}_i + \hat{\sigma}_{j_{mean}} \boldsymbol{\mu}_j, \tag{3}$$

$\boldsymbol{\mu}_i$ and $\boldsymbol{\mu}_j$ represent embedding features of the two images, $\hat{\sigma}_{i_{mean}}$ and $\hat{\sigma}_{j_{mean}}$ represent their corresponding uncertainty values. We introduce negligible computation overhead as we only mix facial features according to their learned uncertainty values in the training phase. Furthermore, the mix feature part will be removed for deployment. Therefore, our network is end-to-end and adds no extra cost on inference.

**Classification Loss** We design an add-up loss to encourage the uncertainty learning branch to learn different uncertainty values for different facial images. Specifically, add-up loss requires the FER model to recognize two expressions of input images simultaneously from mixed features. Since $\tilde{\boldsymbol{\mu}}$ is the mixed feature, we feed $\tilde{\boldsymbol{\mu}}$ to the classifier:

$$L_{total} = -\frac{1}{N} \sum_{i,j}^{N} (\log \frac{e^{\boldsymbol{W}_{y_i} \tilde{\boldsymbol{\mu}}}}{\sum_c^C e^{\boldsymbol{W}_c \tilde{\boldsymbol{\mu}}}} + \log \frac{e^{\boldsymbol{W}_{y_j} \tilde{\boldsymbol{\mu}}}}{\sum_c^C e^{\boldsymbol{W}_c \tilde{\boldsymbol{\mu}}}}). \tag{4}$$

The loss function adds up the loss of recognizing $\tilde{\boldsymbol{\mu}}$ as class $i$ and as class $j$. $y_i$, $y_i$ means $label_i$, $label_j$, $\boldsymbol{W}_c$ is the c-th classifier and $C$ means the total number of expression classes. The loss function forces the model to recognize two expressions equally from the mixed features. When mixing two facial features, there will be a relatively easy facial feature for expression recognition, and the other will be relatively hard. For simplicity, we note $\boldsymbol{\mu}_i$ as the easy facial feature. Our model will learn a small uncertainty value for $\boldsymbol{\mu}_i$ to mix with $\boldsymbol{\mu}_j$ as $\boldsymbol{\mu}_i$ is easy to recognize, the model can already get a small classification loss with its label $y_i$ after training for several epochs, even $\boldsymbol{\mu}_i$ only takes up a small amount of the mixed feature $\tilde{\boldsymbol{\mu}}$. While for the hard image $j$, it needs to take up a large amount of the mixed feature $\tilde{\boldsymbol{\mu}}$ to enable the model to find useful feature to get a small classification loss with label $y_j$.

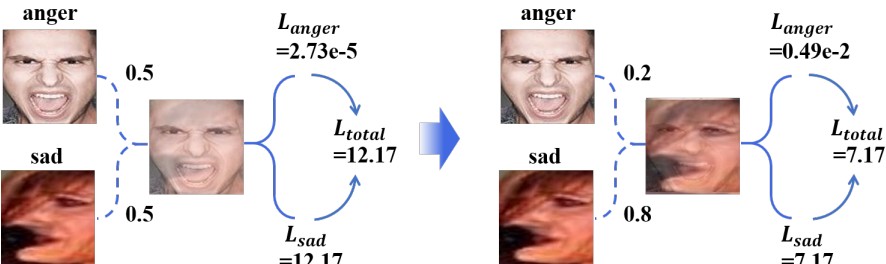

Figure 2: An experiment to show why RUL is effective in uncertainty learning. We mix different features with different weights and print the losses of recognizing the mixed feature to different labels. Visually, the face with anger expression is the certain one, and the face with sad expression is the uncertain one. In the process of feature mixup, RUL will assign a larger weight to the uncertain one to minimize the total loss.

## 3.3 Why RUL is Effective

In this section, we illustrate why RUL is effective in uncertainty learning. For visualization simplicity, we plot the mixed images instead of the mixed features in Figure 2. At the start of the training, RUL might learn similar uncertainty values for all images, which is shown in the left part of Figure 2. As the certain facial expression image with anger label in Figure 2 can be easily recognized from the mixed feature, the model can already get a small loss value with anger label while gets a large loss value with a sad label. In order to get the minimum total loss, the model needs to recognize the sad expression from the mixed feature more effectively. As training progress, the mix weight (uncertainty value) of the uncertain sad image will increase to enable the model to recognize the sad expression from the mixed feature. Thus, RUL will learn larger uncertainty values for uncertain images.

## 4 Experiments

In this section, we first describe three in-the-wild FER datasets and our implementation details. We then evaluate our proposed method on these datasets. We compare our method with state-of-the-art uncertainty learning methods on the accuracy, noisy label training, and accuracy with rejection [41]. Furthermore, we provide qualitative and quantitative analysis to show what is the meaning of our learned uncertainty values. Last, we compare our method to other state-of-the-art FER methods and also demonstrate that RUL is applicable to datasets in different fields. We also carry out an ablation study to find the effectiveness of the mix part in RUL.

## 4.1 Datasets

To evaluate the effectiveness of our model, we select three widely used in-the-wild FER datasets. Compared with FER datasets collected in the laboratory, they have more noises like noisy labels and ambiguous expressions. So they are more complex, variable, and challenging for expression recognition tasks. The details are as follows.

**RAF-DB** [27] is a crowdsourced facial expression dataset that contains 29672 facial images annotated with basic or compound expressions by 40 trained annotators. In this work, we use facial images with 6 basic expressions [8] and neutral expression, including 12271 images as training data and 3068 images as test data.

**FER2013** [14] consists of 35,887 grayscale 48x48 pixel images in total, with 28,709 training samples, 3,589 public test samples, and 3,589 private test samples. These facial images are classified into 7 emotion classes, same as RAF-DB.

**AffectNet** [33] is currently the largest FER dataset, including 440,000 images. The images are collected from the Internet by querying the major search engines with 1,250 emotion-related keywords. There are around 280,000 training images and 4000 testing images annotated by human. They are annotated to 8 emotion classes (7 emotion classes same as RAF-DB and Contempt class).

## 4.2 Implementation Details

Given a batch of images from RAF-DB, we first resize them to 224×224 pixels and use ResNet-18 [18], which is pretrained on Ms-Celeb-1M [17] as backbone. The two branches have the same structure as BatchNorm2d [20]-Dropout [38]-Flatten-FC-BatchNorm1d. We set dropout rate as 0.4, output dimension as 64. Note that the second branch outputs $\log \sigma$ instead of $\sigma$ in order to stabilize the training following [4]. The model is trained in an end-to-end manner with a single GTX 1080ti GPU for 70 epochs with batch size of 64. We also utilize an Adam optimizer [24] with weight decay of 0.0001. The learning rate is initialized as 0.0002 except the last fully connected layer for classification, which is 0.002. We use ExponentialLR [30] learning rate scheduler with gamma of 0.9 to decrease the learning rate after each epoch. The setting under AffectNet dataset is the same as RAF-DB. As for FER2013, we use the same setting as [49] as the image in FER2013 is 48x48 pixel grayscale image.

## 4.3 Evaluation on Label Noises

Both ambiguous facial expressions and inconsistent annotations will cause label noises in FER datasets. We carry out extensive experiments to demonstrate the improvement of RUL compared with state-of-the-art noise-tolerant FER methods. Following SCN [47], we randomly choose 10%, 20%, 30% of training data and flip their labels to other expression categories randomly. To make a fair comparison with SCN, we also consider the most uncertain samples in the training process as samples with label noises and if the maximum prediction probability is higher than the one of the given label with a threshold (set to 0.2 in all cases), we choose to rely on the FER model and change the label to the index of the maximum prediction probability.

We run all experiments three times and compute the mean and standard variance of the results. As shown in Table 1, RUL outperforms SCN and DUL under all circumstances. RUL improves test accuracy by 4.30%, 4.32%, 4.60% with noise ratio of 10%, 20% and 30% on RAF-DB, 1.01%, 1.56%, 2.01% on FER2013 and 1.94%, 2.01%, 4.05% on AffectNet compared with SCN. Experiment results show that the benefit from RUL becomes more obvious as the noise ratio increases up which means RUL is more robust to label noises. Note that we do not use pretrained model when training in FER2013 to find how the pretrained model affects the experiment results. From the results we can draw the conclusion that using pretrained model can better deal with label noise as RUL improves the accuracy most when using pretrained models which aligns to the finding in [19]. The results also demonstrate that RUL can learn meaningful uncertainty values as we use uncertainty values to guide the model to find the most uncertain training samples and suppress label noises.

Table 1: Test accuracy (%) on RAF-DB, FER2013 and AffectNet with synthetic noisy labels.

| Method | Noisy (%) | RAF-DB | FER2013 | AffectNet |
|---|---|---|---|---|
| Baseline | 10 | 80.43±0.72 | 69.25±0.18 | 56.85±0.14 |
| SCN [47] | 10 | 81.92±0.69 | 69.28±0.05 | 58.72±0.20 |
| DUL [4] | 10 | 85.08±0.21 | 69.43±0.27 | 58.26±0.10 |
| RUL | 10 | **86.22±0.29** | **70.29±0.28** | **60.66±0.13** |
| Baseline | 20 | 78.01±0.29 | 64.87±0.32 | 54.74±0.62 |
| SCN [47] | 20 | 80.02±0.32 | 66.30±0.49 | 56.35±0.61 |
| DUL [4] | 20 | 81.95±0.32 | 65.55±0.31 | 56.25±0.09 |
| RUL | 20 | **84.34±0.29** | **67.86±0.35** | **58.36±0.28** |
| Baseline | 30 | 75.12±0.78 | 62.52±0.56 | 51.46±0.52 |
| SCN [47] | 30 | 77.46±0.64 | 62.61±0.82 | 52.60±0.86 |
| DUL [4] | 30 | 78.90±0.80 | 60.98±0.34 | 55.09±0.32 |
| RUL | 30 | **82.06±0.44** | **64.62±0.39** | **56.65±0.13** |

Table 2: Test accuracy (%) versus rejection rate results of different uncertainty learning methods on RAF-DB and FER2013. The best performance among each column is shown in bold form.

| Method | RAF-DB | | | | FER2013 Private Testset | | | |
|---|---|---|---|---|---|---|---|---|
| | 0 rejection | 10% rejection | 20% rejection | 30% rejection | 0 rejection | 10% rejection | 20% rejection | 30% rejection |
| SCN [47] | 87.35 | 86.85 | 86.63 | 87.28 | 72.67 | 72.82 | 74.29 | 75.80 |
| CONF [6] | 88.30 | 91.78 | 93.97 | 95.90 | 72.69 | 73.00 | 73.11 | 72.97 |
| DUL [4] | 88.04 | 90.11 | 92.58 | 94.50 | 73.67 | 76.01 | 79.10 | 82.25 |
| RUL | **88.98** | **92.72** | **95.40** | **97.35** | **73.75** | **77.24** | **80.98** | **84.75** |

## 4.4 Test with Rejection.

In order to illustrate the quality of our learned uncertainty values more intuitively, we utilize an evaluation metric similar to [41, 2, 15, 40] named accuracy versus rejection rate. The metric shows a test accuracy over the fraction of unconsidered facial expression images. Specifically, based on the predicted uncertainty values, these unconsidered images are the most uncertain images to recognize, and the test accuracy is calculated on the remaining images. The metric indicates good uncertainty estimation when the test accuracy increases consistently when increasing the ratio of unconsidered images. We also compare with the out-of-distribution detection method [6] noted as CONF. CONF uses labels as hints to help indicate when the network tends to give wrong predictions. We show the test accuracy in Table 2.We also plot the accuracy versus rejection rate curve in the supplementary material. RUL gets the best performance in all cases, which illustrates that the uncertainty values learned by RUL are more related to the recognition confidence than other methods. Another interesting finding is that CONF performs pretty well on RAF-DB while degrades on FER2013. It may be explained by the following reason that we use pretrained model on RAF-DB while training from scratch on FER2013. CONF needs a strong backbone to make good use of the information from labels while our method can perform well in both ways.

## 4.5 Understand learned uncertainty values.

In this section, we visualize the learned features of different uncertainty learning methods and provide qualitative and quantitative analysis to show the meaning of our learned uncertainty values.

**Visualization of learned features.** We use t-SNE [44] to visualize the learned feature distributions of different uncertainty learning methods to show the effectiveness of RUL. The results are shown in Figure 3. It is shown that the comparison of different facial expressions encourages intra-class compactness and inter-class seperability of the learned features. We believe that this is because RUL needs to recognize both expressions from the mixed feature, and it will be forced to learn the most discriminative feature that can tell an expression image apart from all the other expression images compared with it. We plot the uncertainty distribution of images with high uncertainty and images with low uncertainty seperately in the supplementary material. The results show that the samples with high uncertainty learned by RUL congregate at the center of the figure. They tend to contain ambiguous expressions like neutral (class 6), which can be easily confused with other expressions. However, the samples with small uncertainty are far from each other which are very easy to be recognized. The feature visualization demonstrates that RUL can learn feature distribution which reflects the uncertainty distribution of the dataset.

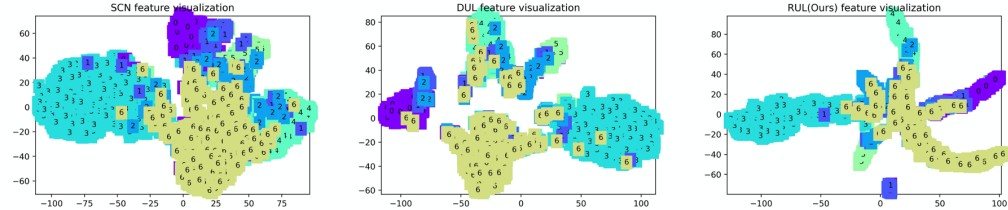

Figure 3: The learned feature distribution of SCN, DUL and RUL on RAF-DB dataset.(0:Surprise, 1:Fear, 2:Disgust, 3:Happy, 4:Sad, 5:Angry, 6:Neutral)

**Visualization of uncertainty values.** We display some expression images in testset with their estimated uncertainty values in Figure 4. The first row shows that images with large uncertainty

values are more likely to contain ambiguous expressions. They are easy to be misclassified (The wrong predicted labels are marked at the bottom of the first row images in red color). In contrast, the second row shows that images with small uncertainty values tend to contain obvious facial features, and they are easy to be rightly classified. This illustrates that our learned uncertainty values can represent the difficulty of expression images as well as the model's confidence of its predictions. That property brings us benefits that are important for the development of safe AI systems. If the model knows when it is prone to be wrong then it can turn to human for help, which might avoid disasters in fields with high accuracy requirements.

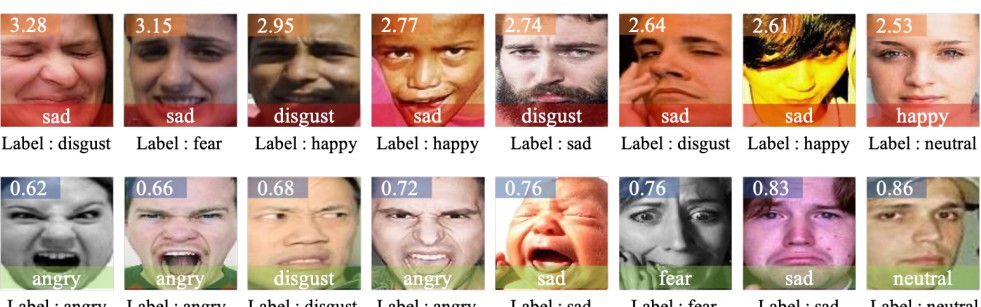

Figure 4: The learned uncertainty values for different images. Images with large and small uncertainty values are shown in the first and second rows respectively. Uncertainty values are marked in the upper left corner, and predicted labels are marked at the bottom. *Note that RUL learns large uncertainty values for images with ambiguous expressions while small uncertainty values for images with obvious expressions.*

**Distribution of uncertainty values.** RUL can learn different uncertainty distributions based on different datasets instead of constraining uncertainty distributions of all datasets to the diagonal multivariate normal Gaussian distribution like DUL. We show the learned uncertainty distributions of different datasets in Figure 5 and Figure 6. In order to make a comparison, we normalize RUL learned uncertainty values to [0,1]. Note that larger uncertainty values mean the images are more uncertain. We cut off at 400 on the y-axis in Figure 5 as SCN learns too many uncertainty values close to 0, which indicates bad performance. The results demonstrate that RUL learns large uncertainty values for more samples while the other two methods learn relative small uncertainty values (smaller than 0.4 on RAF-DB and smaller than 0.6 on FER2013) for all samples, which means RUL can reduce the negative influence from the strong learning ability of networks mentioned in Section 3.1. Furthermore, we also randomly sample several images which are hard for human to recognize and find that only RUL learns large uncertainty values for them.

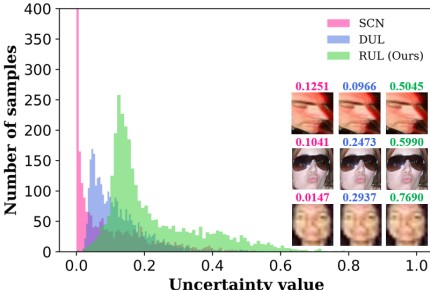

Figure 5: The learned uncertainty distribution of RAF-DB. *Uncertainty values are marked at the top of the image, different colors represent different methods.*

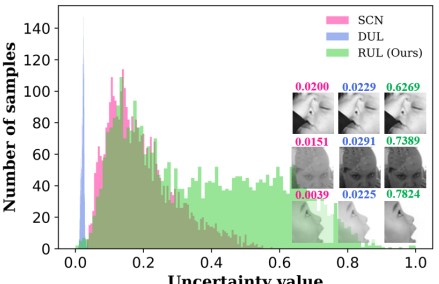

Figure 6: The learned uncertainty distribution of FER2013. *Uncertainty values are marked at the top of the image, different colors represent different methods.*

| Table 3: Comparison on RAF-DB | | | Table 4: Comparison on FER2013 | | | Table 5: Comparison of different structures. | |
|---|---|---|---|---|---|---|---|
| Method | Acc.(%) | | Method | Acc.(%) | | Method | Acc.(%) |
| DLP-CNN [27] | 84.22 | | Conv+Inception [32] | 66.40 | | mix output | 88.36 |
| gaCNN [29] | 85.07 | | Bag of Words [21] | 67.40 | | loss attention | 88.75 |
| IPA2LT [52] | 86.77 | | Deep-Emotion [31] | 70.02 | | RUL | **88.98** |
| RAN [48] | 86.90 | | VGG [35] | 72.70 | | Method | 10% rejection |
| SCN [47] | 87.03 | | SCN [47] | 72.67 | | mix output | 92.17 |
| RUL | **88.98** | | RUL | **73.75** | | loss attention | 89.32 |
| | | | | | | RUL | **92.72** |

Table 6: The test accuracy versus rejection rate of different uncertainty learning methods on CIFAR-10 and ImageNet-100.

| Method | CIFAR-10 | | | | ImageNet-100 | | | |
|---|---|---|---|---|---|---|---|---|
| | 0 rejection | 10% rejection | 20% rejection | 30% rejection | 0 rejection | 10% rejection | 20% rejection | 30% rejection |
| CONF [6] | 93.47 | 96.84 | 98.90 | **99.56** | 76.48 | 76.46 | 76.36 | 76.21 |
| SCN [47] | 95.37 | 94.87 | 94.24 | 94.14 | 75.49 | 74.61 | 74.49 | 74.99 |
| DUL [4] | 95.24 | 98.16 | 99.22 | 99.49 | 76.03 | 75.80 | 79.64 | 82.86 |
| RUL | **95.58** | **98.60** | **99.34** | 99.44 | **77.74** | **77.65** | **82.48** | **86.74** |

## 4.6 Comparison with state-of-the-art FER methods.

Although we aim to deal with uncertainty learning problems in FER, our RUL method still gets better test accuracy comparing to a series of state-of-the-art FER methods according to the results in Table 3 and Table 4.

Note that though RUL uses no extra data for co-training, it still achieves better performance. We speculate that this is because the comparison of different FER images in the training process implicitly introduces the data augmentation effect. Meanwhile, as uncertain images will get larger weights to mix with certain images, the FER model will pay more attention to learning the uncertain images after it fits the certain images, which improves the model's ability to recognize difficult facial expressions. We provide the confusion matrix of different methods on RAF-DB in the supplementary material. As mentioned by [23], the FER model tends to make more mistakes in classes with fewer samples, such as disgust and fear. However, the confusion matrices show that RUL increases test accuracy mainly in these two classes, which means RUL can better recognize difficult facial expressions.

We also replace the Gaussian variances which are the learned uncertainty values in DUL with RUL learned uncertainty values to guide the DUL feature learning branch. Though RUL does not constrain the uncertainty distribution to a Gaussian distribution, it still improves the performance of DUL from 88.17% to 88.75%, which means the uncertainty learned by RUL can better reflect the uncertainty distribution of the dataset.

## 4.7 Other Experiments.

**Experiments on CIFAR-10 and ImageNet-100.** To show RUL is effective not only on FER tasks, we also carry out experiments on CIAFR-10 and ImageNet-100. Table 6 shows that RUL could still improve the performance of the deep learning model on other classification tasks. The test accuracy increases along with the rejection of the most uncertain test samples, which illustrates that RUL can still learn meaningful uncertainty values on datasets besides FER tasks.

**Ablation study.** In order to show that the mixed feature part is useful for uncertainty learning, we implement a mix output method and a loss attention method to make a comparison. Specifically, we do not mix facial features in the mix output method. Instead, we classify two images separately and mix the logits according to uncertainty values. Results are shown at Table 5. RUL performs better than mix output in both the test accuracy and accuracy with 10% rejection, reflecting that mixed features may contain more relativity information than mixed logits. Loss attention method means we skip the mix part and use the uncertainty values to directly weight the loss values. Table 5 shows that RUL outperforms the loss attention method, which illustrates that RUL is effective not only because

uncertainty values affect loss values but also due to the interaction of different facial features during the training process.

## 5   Conclusion

In this paper, we propose a novel and effective uncertainty learning method called Relative Uncertainty Learning for uncertainty quantification and noisy label training. Unlike traditional uncertainty learning methods, which use uncertainty values to weigh the loss and then add regularization to force the network to learn uncertainty, we view uncertainty as a relative concept and learn uncertainty through the relativity of different samples. Extensive experiments have shown that RUL can learn meaningful uncertainty values, which reflect the difficulty of samples to be rightly classified. RUL can also help the FER model achieve state-of-the-art performance in both real-world and synthetic noisy FER datasets. We further demonstrate that RUL is also useful in other classification tasks.

## Acknowledgments and Disclosure of Funding

This work was supported by the National Natural Science Foundation of China under Grants No. 61871052

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
