# Relative Uncertainty Learning for Facial Expression Recognition Supplementary Material

## A   Visualization results on MNIST and CIFAR.

We provide visualization results on MNIST and CIFAR to show our uncertainty learning method also works well on datasets besides facial expression recognition (FER) tasks. We plot the most uncertain images according to RUL learned uncertainty values of CIFAR-10 in Figure 1. We utilize red rectangles to mark images that are misclassified and green rectangles to mark images that are rightly classified. It is shown that 53 out of 104 uncertain images are misclassified by the network which indicates our relative uncertainty learning (RUL) can learn large uncertainty values to the hard images which are more likely to be misclassified. To make a contrast, we display the images with the smallest uncertainty values in Figure 2 and the network gets 100% accuracy on these samples. From Figure 1, 2, we can draw the conclusion that the images with large uncertainty values tend to have ambiguous objects which are easily confused with other objects while the images with small uncertainty values tend to have obvious objects.

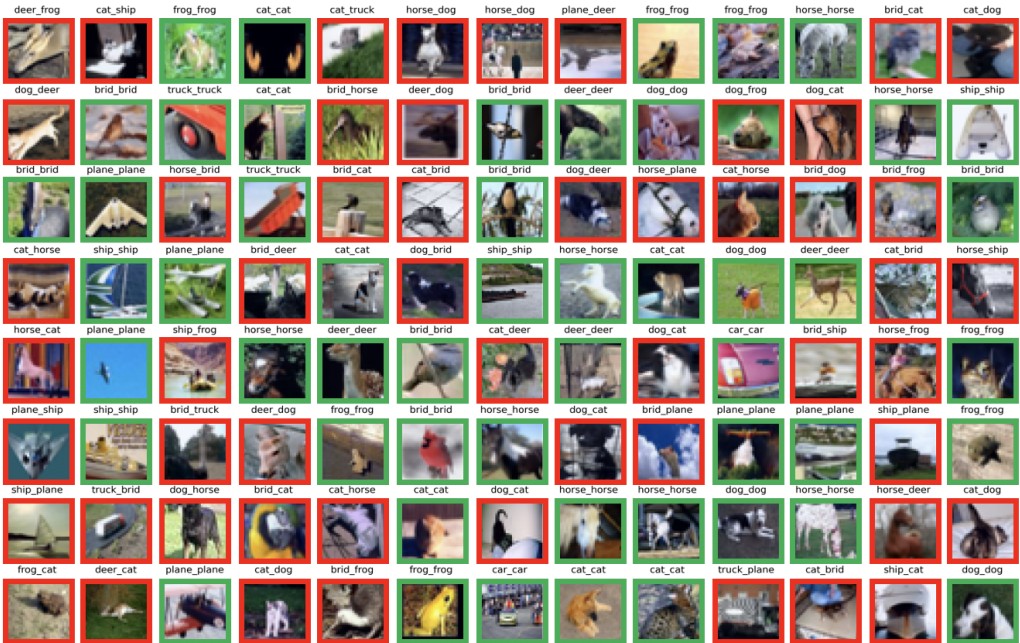

Figure 1: The most uncertain images in CIFAR-10, the label and the predicted label are marked at the top of each image. Red rectangles mean wrong classification while green rectangles mean right classification. *The network only gets 49% accuracy on these most uncertain images.* Better zoom in to view each label of the image.

We also display the most uncertain and certain images of MNIST in Figure 3,4 according to RUL learned uncertainty values. The figures show that the uncertain images are difficult samples that are

35th Conference on Neural Information Processing Systems (NeurIPS 2021).

hard to be rightly classified while the certain images are those easy to be rightly classified. Note that there are lots of number 3 in the certain images, we speculate that might because number 3 has obvious features for classification and is difficult to be confused with other numbers.

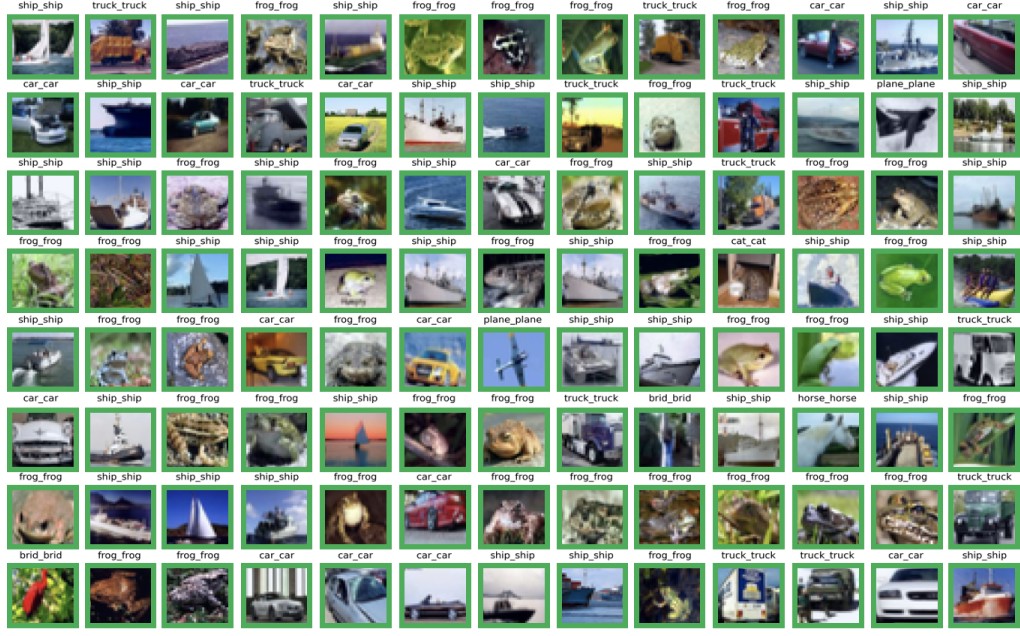

Figure 2: The most certain images in cifar10, the label and the predicted label are marked at the top of each image. Green rectangles mean right classification. *The network gets 100% accuracy on the most certain images.* Better zoom in to view each label of the image.

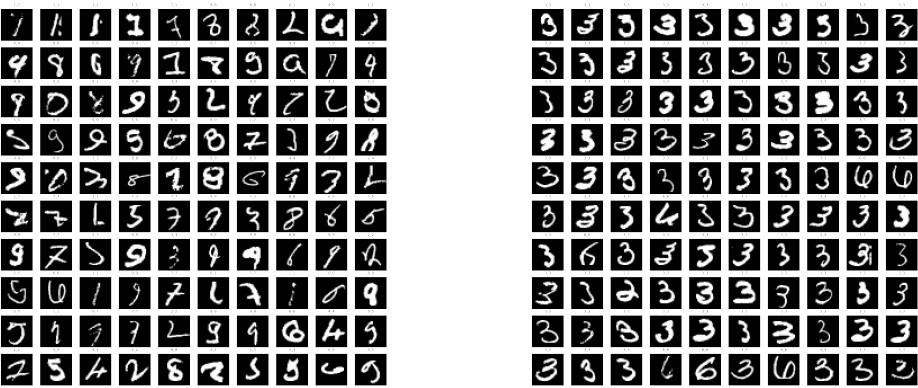

Figure 3: The most uncertain images in MNIST. They are usually very hard to be rightly classified.

Figure 4: The most certain images in MNIST. They are very easy to be rightly classified.

## B  Label noise experiments on MNIST and CIFAR.

We also carry out experiments on MNIST and CIFAR with synthetic noises. The generation of synthetic noises is the same as mentioned in Section 4.3. We train for 35 epochs and start to relabel

Table 1: Test accuracy (%) on MNIST and CIFAR with synthetic noisy labels.

| Method | MNIST | | | | CIFAR | | | |
|---|---|---|---|---|---|---|---|---|
| | 0 noise | 10% noise | 20% noise | 30% noise | 0 noise | 10% noise | 20% noise | 30% noise |
| Baseline | **99.50** | 99.16 | 99.05 | 98.82 | 95.53 | **90.20** | 87.46 | 85.41 |
| SCN | 99.28 | 99.23 | 99.06 | 98.72 | 95.37 | 89.79 | 87.82 | 86.36 |
| DUL | 99.47 | 99.33 | 99.11 | 98.76 | 95.24 | 90.16 | 88.97 | 85.97 |
| RUL (Ours) | 99.45 | **99.44** | **99.42** | **99.36** | **95.58** | 90.19 | **89.60** | **88.21** |

from the 10th epoch in MNIST while 250 epochs and start to relabel from the 100th epoch on CIFAR. According to SCN, relabeling module only considers 50% samples with the largest uncertainty values. If the maximum prediction probability is higher than the one of given label with a threshold (set to 0.2), we believe that sample contains label noise and then change the label to the index of the maximum prediction probability. The results are shown at Table 1. With the increase of the label noise, the performance improvement of RUL is more obvious which means our RUL is more robust to label noise with the help of uncertainty learning.

## C   Visualization of features with large uncertainty and small uncertainty.

Figure 5 shows that features with small uncertainty learned by DUL tend to lie in the centers of emotion classes. We speculate that this is because DUL adds an extra Gaussian regularization. Hence, the features with small uncertainty are those with small std values near the centers of emotion classes. In contrast, the features with large uncertainty are the features with large std values, which lie far from the centers of emotion classes. However, it is shown that the features with large uncertainty values learned by DUL are not always hard to be rightly classified. The features far from the center of a certain class might also be far from other classes, which means that they are unlikely to be misclassified to other classes. When it comes to RUL, things are different. Samples with small uncertainty are well classified and have large inter-class distances, while samples with large uncertainty are in the center of the figure with small inter-class distances reflecting the real data uncertainty distribution.

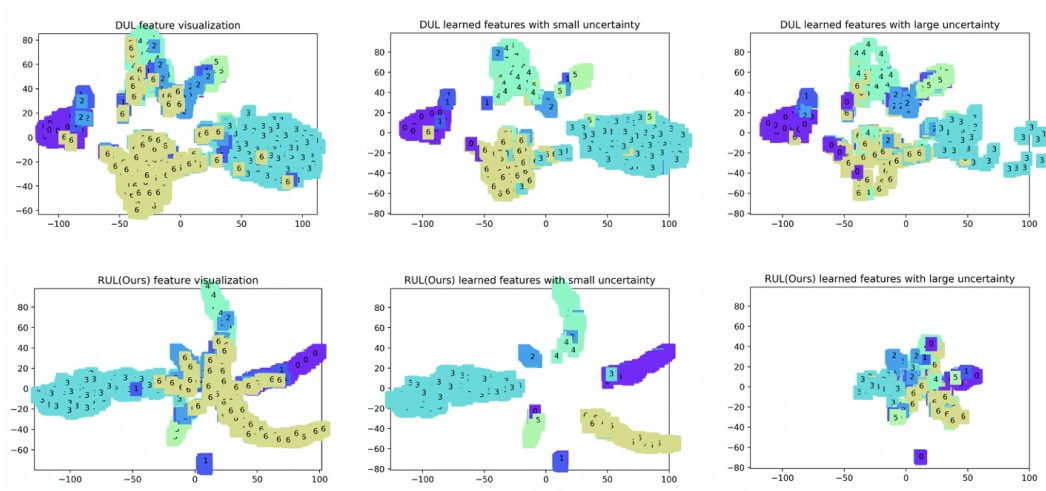

Figure 5: The learned features with large uncertainty and small uncertainty by DUL and RUL.

## D   Visualization of different uncertainty learning methods on RAF-DB.

We provide visualization results of different uncertainty learning methods to explore the property of their learned uncertainty values. Figure 6 shows the most uncertain and certain images according to the uncertainty values learned by SCN. The most uncertain images contain some images with

low quality. However, only 4 of these images being misclassified by the network, which means the learned uncertainty values cannot represent the confidence of the prediction of the model. Note that the most certain images find by SCN are all belong to happy class. As happy is the class with most training samples and highest accuracy in all 7 expression classes which indicates SCN cannot provide more useful information in finding samples with low uncertainty. Shown by Figure 7, DUL can better indicate the confidence of the prediction as 26/100 of the most uncertain images are misclassified by the network. However, most of the certain images find by DUL are also belong to happy class. Our uncertainty learning method can learn uncertainty values which can better reflect the confidence of the prediction as 55/100 of the most uncertain samples find by RUL are misclassified by the network. Furthermore, the most certain images find by RUL are not confined to the easiest class happy. Instead, they usually contain obvious expression features which make them easy to be rightly classified.

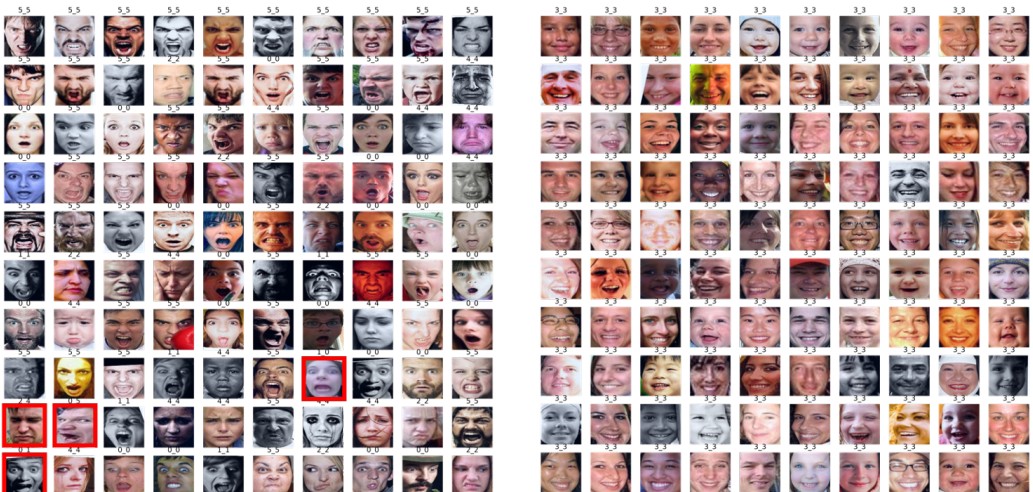

Figure 6: The most uncertain and certain images according to the uncertainty values learned by SCN. *There are only 4/100 of the most uncertain samples being misclassified by the network. 100/100 of the most certain samples being rightly classified by the network.* Note that the most certain images all belong to happy class.

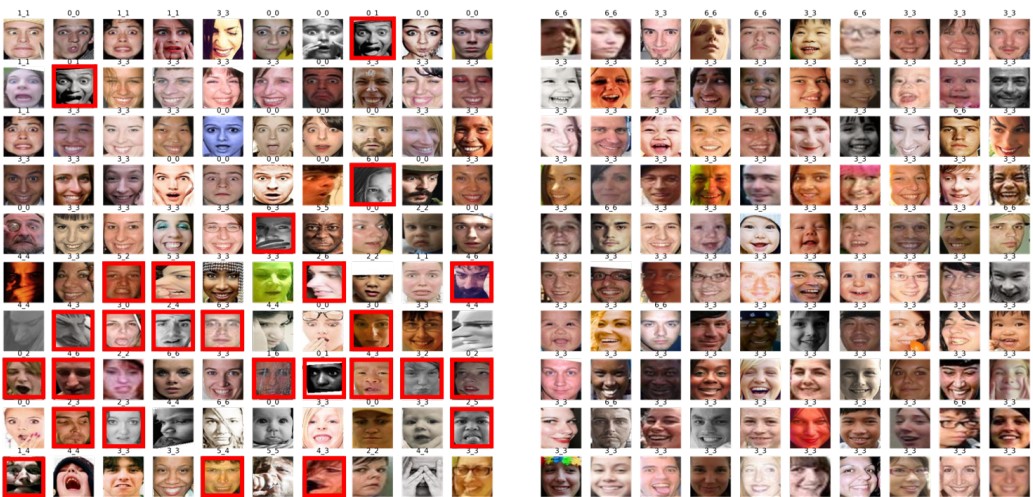

Figure 7: The most uncertain and certain images according to the uncertainty values learned by DUL. *There are 26/100 of the most uncertain samples being misclassified by the network. 100/100 of the most certain samples being rightly classified by the network.*

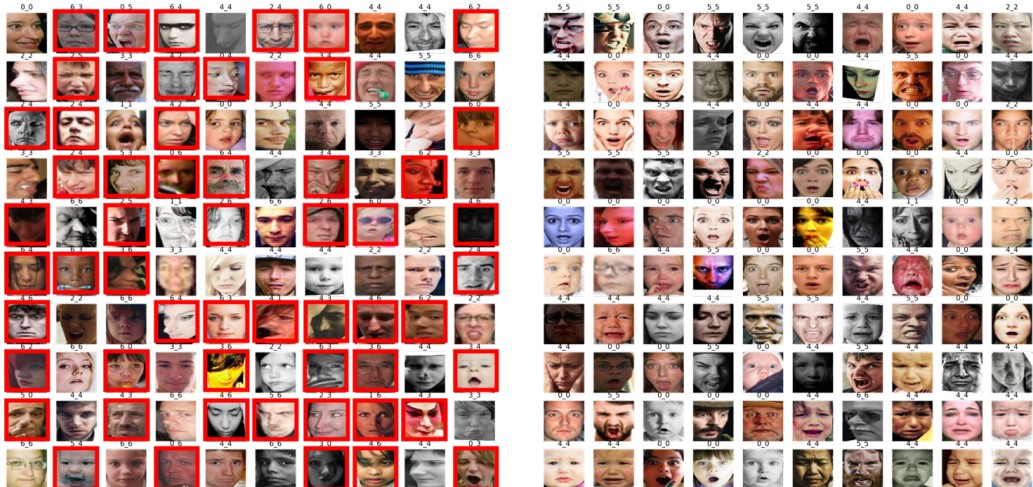

Figure 8: The most uncertain and certain images according to the uncertainty values learned by RUL (Ours). *There are 55/100 of the most uncertain samples being misclassified by the network. 100/100 of the most certain samples being rightly classified by the network.*

# E   FER2013 implementation details.

We provide the implementation details of experiments on FER2013 mentioned in Section 4.2. In order to verify the effectiveness of RUL when training from scratch, we carry out experiments on FER2013 without using pretrained ResNet as backbone. Given a batch of images from FER2013, we first crop them into 44x44 size and then utilize random horizontal flip. In the test phase, we use the TenCrop function with crop size 44x44 and average the ten outputs to get the predicted logits. We utilize SGD optimizer with learning rate initialized as 0.01, momentum as 0.9 and weight decay as 5e-4. We train for 250 epochs with batch size 128. We also decay learning rate from the 80 epoch every 5 epochs with the decay rate as 0.9.

# F   The accuracy versus rejection rate curve.

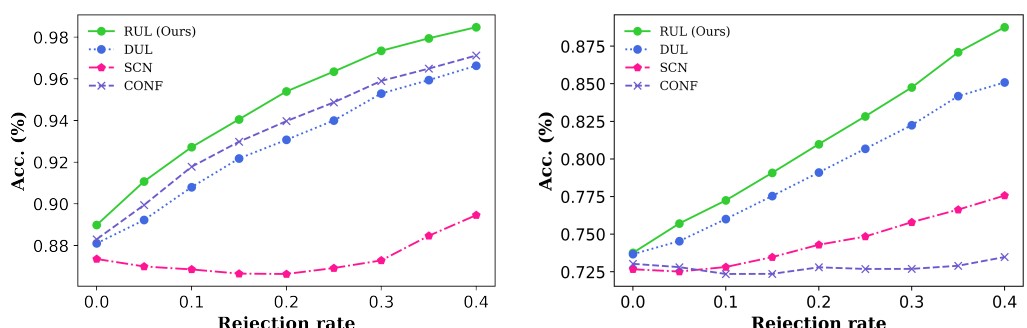

Figure 9: Accuracy versus rejection rate curves on RAF-DB.

Figure 10: Accuracy versus rejection rate curves on FER2013 Private Testset.

# G   The details of mix output method and loss attention method.

We provide more details of mix output and loss attention methods mentioned in Section 4.7. RUL mixes two different features and then encourages the model to recognize both expressions from the mixed feature while mix output method mixes the logits which are the outputs of the classifier and then uses the mixed logit to compute losses with two labels of the input images. We argue that this

experiment can illustrate the features contain more information than the logits as RUL outperforms mix output method. The structure of loss attention method is shown in Figure 12. In order to show that the effectiveness of RUL is not only comes from weighting different losses, we implement loss attention method. We skip the mix feature part of RUL and compute losses using features and their corresponding labels. We then get the total loss by adding two different losses according to their uncertainty values. RUL outperforms loss attention method which means that RUL is effective not only because the regularization effect of the add-up loss but also due to the interaction of different features in the training process.

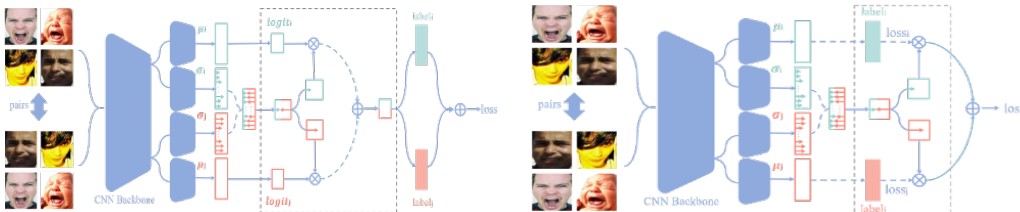

Figure 11: The pipeline of mix output method. The different structure from RUL is marked with a grey dotted rectangle.

Figure 12: The pipeline of loss attention method. The different structure from RUL is marked with a grey dotted rectangle.

## H    The confusion matrices of different uncertainty learning methods on RAF-DB.

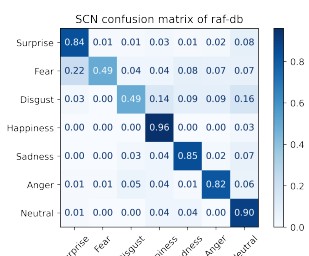 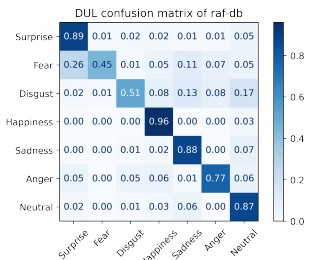 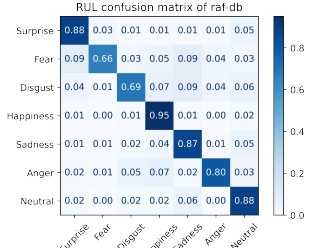

Figure 13: SCN confusion matrix on RAF-DB.

Figure 14: DUL confusion matrix on RAF-DB.

Figure 15: RUL confusion matrix on RAF-DB.

## I    Ablation study.

We carry out an ablation study to find out the suitable dimension of the mixed features for FER. It is shown in Figure 16 that RUL gets the best performance with feature dimension 32 in both accuracy and accuracy with 10% rejection, but RUL is not sensitive to different feature dimension values, which means RUL can get good recognition performance with little or no hyperparameter tune. We also provide results of different mix numbers in Figure 17. The experiments show that we get the best performance when we mix two images together in the training process. The performance drops when we mix five images together and we speculate the reason might be that mixing too many images together increases the difficulty of finding the relativity of different images. The x-axis C(3,2) means we choose three images and mix them in pairs while x-axis 3 means we mix three images together. Figure 17 shows that simply mixing images in pairs from the three input images works better than mixing three images together.

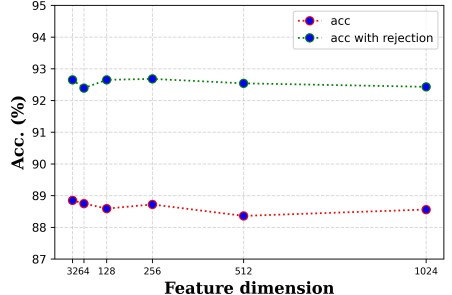
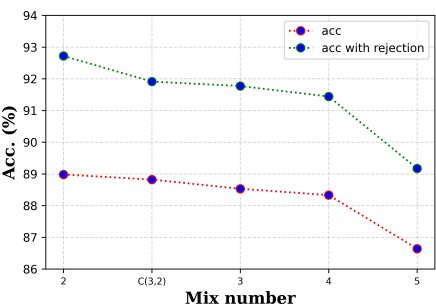

Figure 16: Different feature dimension with accuracy and accuracy with rejection.

Figure 17: Different mix number with accuracy and accuracy with rejection.