# OpenReview forum: "Relative Uncertainty Learning for Facial Expression Recognition"
_NeurIPS.cc/2021/Conference — NeurIPS 2021 Poster_

### Official Review · Reviewer_3dyN · 2021-07-12

**Rating:** 6
**Confidence:** 4

**Summary:**

This paper focuses on the uncertainty problem in the facial expression recognition. To quantify the uncertainties and improve the FER performance, this paper proposes the relative uncertainty learning (RUL), which could learn the uncertainty from the relative difficulty. Experiments on two public FER datasets show the effectiveness of the proposed method.


**Limitations And Societal Impact:**

Yes

**Main Review:**

Pro:

+Considering the relative idea and human cognition, the RUL learns the uncertainty though comparisons.

+Experiments on the basic RAF-DB are encouraging.

Cons:

-The novelty of this work is not enough for NeurIPS acceptance standards. And the add-up loss could be regarded as the cross-entropy loss, which needs to be calculated twice.

-The experiments of this paper are insufficient. Especially, the RUL is just evaluated on RAF-DB and FER-2013. The latter is a disputed FER dataset, and is not widely used at present.

-English needs a good revision in some parts of the text, which also makes it difficult to grasp some details.


### Comment

1) The defination of the facial expression pairs is not clear. How to ensure that the image i and its counterpart image j belong to different expression categories?

2) Are the weights of branches model for image i and j sharing? Besides, the output demension D is not given, which is an important parameter in this work.

3) The designed normalized uncertainty vectors are averaged following DUL [1]. What happens if this branch maps faical features to a uncertainty probalibility via the softmax function? It would be better to further discuss more ways.

4) The add-up loss can be viewed as the two-time cross-entropy loss. Why the sum is divided by N, not 2N? In addition, the explanation below the Equ.4 is wrong. It should be 'y_{i}, y_{j} means label_{i}, label_{j}'.

5) The generalization performance of RUL is not convincing. It would be better to evaluate the proposed RUL on more FER datasets, especially AffectNet [2] and FERPlus [3]. The authors only compare with some common methods on FER-2013, not with uncertainty learning methods, for example the SCN [4].


[1] Data uncertainty learning in face recognition. CVPR2020.
[2] Affectnet: A database for facial expression, valence, and arousal computing in the wild. IEEE Trans. on Affective Computing 2017.
[3] Training deep networks for facial expression recognition with crowd-sourced label distribution. ACM ICMI2016.
[4] Suppressing Uncertainties for Large-Scale Facial Expression Recognition. CVPR2020.


After the rebuttal, I decided to raise my score to weakly accept.

**Time Spent Reviewing:**

2

---

> ### Author Response · Authors · 2021-08-10
> **Response to Reviewer 3dyN's comments**
>
> Thank you for your time and effort spent on reviewing our paper. Your thorough main review and comments are very important to the improvement of our work, which we address below:
>
> **Cons1.** Thank you for bringing up the question about the novelty of our work. Though the methods used in our work are not new which are manifold mixup and cross-entropy loss, our idea is well-grounded and motivated as we try to use manifold mixup and cross-entropy loss to simulate human cognition—uncertainty comes from comparison. It is novel that mixup weights can be learnable and represent the uncertainty of input images. We are also the first to view uncertainty from a relative concept and leverage a mixup framework for uncertainty learning. The add-up loss is also used as a regularization for uncertainty learning for the first time to enable the uncertainty learning ability from comparison.
>
> **Cons2.** Thank you for your suggestion. We carried out experiments on AffectNet [1] and FERPlus [2] as suggested. The details are shown in the response of **Comment5.**
>
> **Cons3.** As suggested, we did a thorough writing review and polished our manuscript to make sure that readers can quickly grasp the details of our paper.
>
> **Comment1.** Thanks for your suggestion. We neglected to mention the code implementation detail of how to enable image i and image j to have different labels in our paper. As mentioned in our paper line 126, "First, we mix a mini-batch with their shuffled ones," we sample image j which has a different label with image i from the same mini-batch. The codes are shown below with annotation:
>
> ```
> new_idx = [ ]	#the indexes to sample input batch to get their mix counterparts`
>
> for i in range(batch_size):
>
> ​    random_idx = random.randint(0, batch_size-1)
>
> ​    while( label[random_idx]==label[i] ):	#ensure label of image j != label of image i
>
> ​        random_idx = random.randint(0, batch_size-1)
>
> ​    new_idx.append(random_idx)
>
> index = torch.from_numpy(np.array(new_idx)).cuda()
>
> #we only need to index the input batch to get their counterparts
>
> sigma1 = sigma / (sigma + sigma[index])
>
> sigma2 = sigma [index] / (sigma + sigma[index])
>
> mixed_mu = sigma1 * mu + sigma2 * mu[index, :]
> ```
>
>
>
> **Comment2.** Yes, image i and j share the same weights of branches model. And we have mentioned the tuning of dimension D in our supplementary material, the experiment results show that our method is not sensitive to dimension D. It can be simply set to 32, 64, 128, 256 or 512 et al. As shown in supplementary line 66, ''we carry out an ablation study to find out the suitable dimension of the mixed features for FER. It is shown in Figure 10 that RUL gets the best performance with feature dimension 32 in both accuracy and accuracy with 10% rejection, but RUL is not sensitive to different feature dimension values, which means RUL can get good recognition performance with little or no hyperparameter tune."
>
> **Comment3.** As suggested, we discussed more ways to map features to uncertainty values. The experiment results are shown below.
>
> | Method  | acc.  | acc. with 10% rejection | acc. with 20% rejection | acc. with 30% rejection |
> | :------ | :---: | :---------------------: | :---------------------: | :---------------------: |
> | FC      | 88.40 |          90.51          |          91.81          |          92.50          |
> | Softmax | 88.59 |          92.47          |          94.62          |          96.65          |
> | RUL     | 88.98 |          92.72          |          95.40          |          97.35          |
>
> FC+Sigmoid means we change the output dimension of the uncertainty learning branch to 1 to directly map features into uncertainty values instead of using averaged uncertainty vectors. Softmax means we normalize uncertainty values utilizing softmax function instead of using linear normalization. It is shown that our method performs the best. FC does not perform well in uncertainty learning which might be because the model neglects some useful information as the output dimension of the uncertainty learning branch is changed to 1 which is much smaller than the former dimension D. Softmax is inferior to our method which might because softmax is an exponential function which makes the difficult images have much larger uncertainty values and the easy images have much smaller uncertainty values thus making it difficult for add-loss optimization.
>
> **Comment4.** The sum can be divided by N or 2N, as this only effects the value of the loss by a constant factor which can be balanced with different learning rates. Thank you for pointing out a typo for us, we modified the misspelling in our manuscript as suggested.
>
> **Comment5.** We ran additional experiments on AffectNet [1] and FERPlus [2] comparing with uncertainty learning method SCN [3] as suggested. As shown in below, our method outperforms SCN in all experiment settings. We will include the experiment results in our final paper.
>
> | Pretrain | Noise (%) | Method  | AffectNet |  FERPlus  |
> | :------: | :-------: | :-----: | :-------: | :-------: |
> |    x     |     0     | SCN [3] |   47.28   |   83.42   |
> |    x     |     0     |   RUL   | **49.60** | **85.21** |
> |    x     |    10     | SCN [3] |   46.29   |   78.83   |
> |    x     |    10     |   RUL   | **48.53** | **80.62** |
> |    x     |    20     | SCN [3] |   42.46   |   74.16   |
> |    x     |    20     |   RUL   | **46.04** | **78.96** |
> |    x     |    30     | SCN [3] |   40.89   |   72.71   |
> |    x     |    30     |   RUL   | **44.77** | **76.92** |
> |    √     |     0     | SCN [3] |   60.23   |   88.01   |
> |    √     |     0     |   RUL   | **61.43** | **88.75** |
> |    √     |    10     | SCN [3] |   58.60   |   84.99   |
> |    √     |    10     |   RUL   | **60.54** | **86.93** |
> |    √     |    20     | SCN [3] |   57.51   |   83.35   |
> |    √     |    20     |   RUL   | **59.01** | **85.05** |
> |    √     |    30     | SCN [3] |   54.60   |   82.20   |
> |    √     |    30     |   RUL   | **56.93** | **83.90** |
>
> As FERPlus has the same data with FER-2013 just with refined labels and the SCN paper provides results on FERPlus, we believe that the experiments above are fair enough and have shown the superiority of our method.
>
>
>
> [1] Ali Mollahosseini, Behzad Hasani, Mohammad H Mahoor, and Mohammad H Mahoor. Affectnet: A database for facial expression, valence, and arousal computing in the wild. TAC, 10(1):18–31, 2017.
>
> [2] Emad Barsoum, Cha Zhang, Cristian Canton Ferrer, and Zhengyou Zhang. Training deep networks for facial expression recognition with crowd-sourced label distribution. In ACM ICMI, 2016.
>
> [3] K. Wang, X. Peng, J. Yang, S. Lu, and Y. Qiao. Suppressing uncertainties for large-scale facial expression recognition. In Proceedings of the IEEE/CVF Conference on Computer Vision and Pattern Recognition, pages 6897–6906, 2020.

---

> > ### Comment · Reviewer_3dyN · 2021-08-23
> > **The new results look good**
> >
> > Most concerns have been addressed. However, there are several problems that have not been solved. First, I think the sum in add-up loss should be divided by 2N. As for the explanation about N or 2N, I think the reason still lacks support. Moreover, the authors should focus on the strictness of the formula. Second, the additional experiments are not sufficient. Like I said in the original comments, the authors should compare uncertainty learning methods on FER-2013.

---

> > > ### Author Response · Authors · 2021-08-24
> > > **Thanks for your kind feedback**
> > >
> > > Thanks very much for your kind feedback. We address the concerns as follows.
> > >
> > > **Concern1.** First, we acknowledge that the traditional mixup [1] method divides the loss by 2N instead of N. However, multiplying the loss with a constant factor 2 does not affect the convergence of the loss function. We can always change the learning rate to get similar performance. Furthermore, we think that the key of our method is to learn meaningful uncertainty values. Therefore, we should not be trapped by whether the loss function should be divided by 2N or N. It is the relative scale of the two cross-entropy losses that really matters. It is the same whether the weights of the losses are 1:1 (divided by N) or 0.5:0.5 (divided by 2N).
> > >
> > > To make our statement more convincing, we further conduct experiments. We take the state-of-the-art performance on RAF-DB as an example. We just divide the loss by 2N and multiply the learning rate by a factor of 2 with no further hyper-parameter tuning, the results are shown below:
> > >
> > > | Method            | 0 rejection | 10% rejection | 20% rejection | 30% rejection |
> > > | ----------------- | :---------: | :----------: | :----------: | :----------: |
> > > | RUL               |  **88.98**  |  **92.72**   |  **95.40**   |  **97.35**   |
> > > | loss divided by 2N |    88.85    |    92.32     |    94.91     |    96.65     |
> > >
> > >  The tiny difference in the performance of models trained with different strategies might be caused by the momentum of the optimizer.
> > >
> > > We believe this is a pure hyper-parameter tuning question. Since RUL gets the better performance, we will keep the loss divided by N. We have also checked the formulas in our paper and would make sure that they are strict in our final paper.
> > >
> > > **Concern2.** We have already compared with uncertainty learning methods like SCN [2] and DUL [3] on FER2013 [4] thoroughly in our paper, the results are shown in Table 2. The 0 rejection column of Table 2 in our paper has already shown the comparison results with SCN and DUL on FER2013, so we do not provide the results again in Table 4, which might cause your concern. We summarize the results in our paper as below.
> > >
> > > | Method             | Acc. (%)  |
> > > | :----------------- | :-------: |
> > > | GoogleNet [5]      |   65.20   |
> > > | Conv+Inception [6] |   66.40   |
> > > | Bag of Words [7]   |   67.40   |
> > > | Deep-Emotion [8]   |   70.02   |
> > > | VGG [9]            |   72.70   |
> > > | SCN [2]            |   72.69   |
> > > | DUL [3]            |   73.67   |
> > > | RUL                | **73.75** |
> > >
> > > | Method | Noise (%) | FER2013 Acc. (%) |
> > > | ------ | :-------: | :--------------: |
> > > | SCN [2]    |    10     |    69.28±0.05    |
> > > | DUL [3]   |    10     |    69.43±0.27    |
> > > | RUL    |    10     |  **70.29±0.28**  |
> > > | SCN [2]    |    20     |    66.30±0.49    |
> > > | DUL [3]    |    20     |    65.55±0.31    |
> > > | RUL    |    20     |  **67.86±0.35**  |
> > > | SCN [2]    |    30     |    62.61±0.82    |
> > > | DUL [3]    |    30     |    60.98±0.34    |
> > > | RUL    |    30     |  **64.62±0.39**  |
> > >
> > >
> > > | Method | Rejection (%) | FER2013 Acc. (%) |
> > > | ------ | :-----------: | :--------------: |
> > > | SCN [2]   |      10       |      72.82       |
> > > | DUL [3]   |      10       |      76.01       |
> > > | RUL    |      10       |    **77.24**     |
> > > | SCN [2]   |      20       |      74.29       |
> > > | DUL [3]   |      20       |      79.10       |
> > > | RUL    |      20       |    **80.98**     |
> > > | SCN [2]    |      30       |      75.80       |
> > > | DUL [3]   |      30       |      82.25       |
> > > | RUL    |      30       |    **84.75**     |
> > >
> > >
> > >
> > > If you have any further questions, do feel free to contact us. We will try our best to clarify them.
> > >
> > >
> > >
> > > [1] H. Zhang, M. Cisse, Y. N. Dauphin, and D. Lopez-Paz. mixup: Beyond empirical risk minimization. In International Conference on Learning Representations, 2018.
> > >
> > > [2] K. Wang, X. Peng, J. Yang, S. Lu, and Y. Qiao. Suppressing uncertainties for large-scale facial expression recognition. In Proceedings of the IEEE/CVF Conference on Computer Vision and Pattern Recognition, pages 6897–6906, 2020.
> > >
> > > [3] J. Chang, Z. Lan, C. Cheng, and Y. Wei. Data uncertainty learning in face recognition. In Proceedings of the IEEE/CVF Conference on Computer Vision and Pattern Recognition, pages 5710–5719, 2020.
> > >
> > > [4] I. J. Goodfellow, D. Erhan, P. L. Carrier, A. Courville, M. Mirza, B. Hamner, W. Cukierski, Y. Tang, D. Thaler, D.-H. Lee, et al. Challenges in representation learning: A report on three machine learning contests. In International conference on neural information processing, pages 117–124. Springer, 2013.
> > >
> > > [5] P. Giannopoulos, I. Perikos, and I. Hatzilygeroudis. Deep learning approaches for facial emotion recogni-tion: A case study on fer-2013. In Advances in hybridization of intelligent methods, pages 1–16. Springer, 2018.
> > >
> > > [6] A. Mollahosseini, D. Chan, and M. H. Mahoor. Going deeper in facial expression recognition using deep neural networks. In 2016 IEEE Winter conference on applications of computer vision (WACV), pages 1–10. IEEE, 2016.
> > >
> > > [7] R. T. Ionescu, M. Popescu, and C. Grozea. Local learning to improve bag of visual words model for facial expression recognition. In Workshop on challenges in representation learning, ICML, 2013.
> > >
> > > [8] S. Minaee, M. Minaei, and A. Abdolrashidi. Deep-emotion: Facial expression recognition using attentional convolutional network. Sensors, 21(9):3046, 2021.
> > >
> > > [9] C. Pramerdorfer and M. Kampel. Facial expression recognition using convolutional neural networks: State of the art. CoRR, abs/1612.02903, 2016.

---

> > > > ### Comment · Reviewer_3dyN · 2021-08-25
> > > > **I raise my score**
> > > >
> > > > Hi authors,
> > > > Thanks for your response. My concerns have been essentially addressed. I think this paper is solid. The new results are enough sufficient for this paper to be published. Therefore, I debate raising my score and recommend accepting this paper.
> > > > However, to ensure the integrity of related work, the following work [*] about uncertainty estimation for FER should also be included in the final paper.
> > > > [*] Dive into Ambiguity: Latent Distribution Mining and Pairwise Uncertainty Estimation for Facial Expression Recognition. CVPR2021.

---

> > > > > ### Author Response · Authors · 2021-08-25
> > > > > **Thanks for your positive feedback**
> > > > >
> > > > > Dear reviewer, Thanks for your time and efforts spent on reviewing our paper. As the paper Dive into Ambiguity: Latent Distribution Mining and Pairwise Uncertainty Estimation for Facial Expression Recognition is closely related to our work, we are very glad to include it in our final paper.

---

### Official Review · Reviewer_RFnf · 2021-07-12

**Rating:** 7
**Confidence:** 5

**Summary:**

The paper presents a novel method for uncertainty learning in the context of facial expression recognition. As in many existing methods for uncertainty quantification, the method comprises adding an extra branch in the output layer of a network solely used for that purpose. Contrary to other methods in uncertainty estimation, the learning of the network is done in a pairwise manner, where the uncertainty for a given image is estimated relative to the peer image. The output features before the emotion classification layer are combined using the estimated uncertainty and are sent to the classifier to be simultaneously classified as both the two emotion categories of each image. This way, for the classifier to “pay more attention” to the image that is harder to classify, the relative uncertainty must be larger w.r.t. that of the easy image. Under this motivation, the weights assigned to the feature fusion during the learning stage are representative of the uncertainty the network assigns to a particular image. The method can therefore be directly used in a single-image way at test time, adding no complexity at inference. The method is validated in several datasets for facial expression recognition, showing superiority in the process of uncertainty estimation w.r.t. other methods based on that extra branch being used to parameterize the standard deviation of a Gaussian distribution, which directly relates to the uncertainty estimation.

**Limitations And Societal Impact:**

The paper does not contain a limitations and societal impact section. I think authors could improve the paper in this regard.

**Main Review:**

The idea is simple yet interesting and elegant, and it appears to work well in practice. It is also a well-grounded and motivated idea. The uncertainty-driven mixup approach for uncertainty estimation is novel and might generate impact in the domain of representation learning as well.


While the idea seems to work, the procedure sounds a bit counterintuitive to me. I understand that the pipeline is inspired from the well-established mix-up framework for visual representation learning. However, from my understanding, the feature fusion described in Section 3 is indeed bringing the features closer (in terms of distance), making them hard to be distinguished. Indeed, I would like to ask the authors to include in the rebuttal a t-SNE visualisation of the “mu” features from both their method and that of DUL. If the method is forced to classify both emotions correctly from the same feature combination, then the features might tend to become closer. This goes in the opposite direction of making the features more distinctive of the underlying emotion, assuming these to be mutually exclusive, or even from the research direction that advocates for label distribution learning. If this is the case, it looks to me that the method is expected to put the learning emphasis on the uncertainty value sigma, totally dismissing the capacity of learning meaningful features. Also, in the mixup framework, the target label (one-hot encodings) are also weighted by the same factor that mixes the feature representation. However, in this paper the classifier is expected to classify both emotions equally, which adds more confusion to how this works. I would like to ask authors to discuss about that in the rebuttal.


Why is sigma set to a scalar in Equation 3? I understand that this value (the mean value of sigma) will represent the total uncertainty, but why the weighting of the features is the same for all the dimensions of the feature space? If sigma is of the same dimension as mu, then why not trying a weighting scheme for each dimension? Such an experiment would also help identify if the features are indeed becoming closer or not as I mentioned above: if using a per-dimension weight makes the method work worse then this could mean that the use of a scalar is limiting to which extend the features can become closer (i.e. their relative angle cannot be changed with just a scalar).


I believe that the paper should also include an out-of-distribution experiment, akin to those of mixup papers. In particular, a cross-dataset experiment should be included, in order to measure how well the uncertainty estimation generalizes to other data domains. I also believe that authors should consider newer databases such as AffectNet. Finally, and as a mere suggestion, I believe the paper would greatly benefit from experimenting in more difficult tasks such as valence and arousal, where the uncertainty is more obvious.


The method seems quite generic and could have impact in other domains. While there are some results in MNIST and CIFAR, these datasets are quite outdated and saturated (as the results show). I believe that an experiment in ImageNet (or ImageNet100 should the former demand too many resources).

The writing needs to be polished and proof-reading is necessary. For example:

l. 41 “the difficulty of classification is a person’s subjective feeling (that??) comes from comparison”
l. 61 “In (the) face recognition field”
l. 62 “Because noisy face images are usually out of the cluster and have larger variances in the latent embedding space which might cause wrong recognition [36].” Incomplete sentence?
l. 77 “the datasets for facial expression recognition is (are?) getting larger and larger”.
l. 99 “The uncertainty learning methods mentioned before suffer from the strong learning ability of deep neural networks more or less.” More or less??
l. 135 “ned by DUL [4], (the??) mean of the”
l. 151 “When mixing two facial features, there will be a relatively easy facial feature for expression recognition and the other is relatively hard.” Different use of tenses (one feature will be and the other already is?)
l. 174 “We also carry out (an??) ablation study”

I would recommend the authors to undergo a thorough writing review to bring the quality of the reading to what is expected for a NeurIPS paper.


==== Score updated from 6 to 7 after rebuttal


After balancing the pros and the cons, I am borderline, towards acceptance, with the paper. It contains some novelty and I believe that, if amended properly, the paper can have a larger impact beyond the field emotion recognition. I would like to encourage the authors to address the main concerns in the rebuttal, to open a fruitful discussion with the fellow peers and AC. In any case, it is an interesting paper, and should it be deemed enough to be accepted to the main venue, I would really encourage authors to improve presentation and readability through a thorough proof-reading, that also uses either British or American English, and not both.


**Time Spent Reviewing:**

8

---

> ### Author Response · Authors · 2021-08-10
> **Response to Reviewer RFnf's comments**
>
> Thank you for your time and effort spent on reviewing our paper. Your thorough comments are very important to the improvement of our work, which we address below:
>
> **Bring features closer.** To address your question, we visualized the learned features of DUL [1], SCN [7], and our method using t-SNE [2] as suggested. The results are provided at https://anonymous.4open.science/r/rebuttal2021-6D55/README.md. We plan to include them in our final paper. Instead of making features hard to distinguish, the results show that our method can make the intra-class distance more compact and the inter-class distance more disperse, making the features easier to distinguish.
>
> To better visualize the relation of learned uncertainty and features, we also split t-SNE into two parts according to uncertainty for better visualization. It is clear that our method encourages intra-class compactness and inter-class separability of samples with small uncertainty. We consider this is caused by  the comparison of images with different labels. As RUL needs to recognize both expressions from the mixed feature, it will be forced to learn the most discriminative feature that can tell an expression image apart from all the other expressions. Meanwhile, the samples with high uncertainty congregate at the center point. They tend to contain ambiguous expressions like neutral (class 6), which can be easily confused with other expressions. In conclusion, our method makes the feature distribution more reasonable, not just make the feature closer.
>
> **Classify emotions equally.** We classify emotions equally to form a regularization for uncertainty learning. However, that does not mean RUL can not learn meaningful features. To get total minimum loss, RUL learns larger weights for uncertain images than the traditional mixup framework, which is 0.5. Deep neural networks have been observed to first fit the easy training samples before eventually memorizing the hard examples [3] [4]. After several training epochs, the easy samples have already been fitted well. Thus, still mixing features equally like traditional mixup is not beneficial for the improvement of classification accuracy. Our method gives hard samples larger mix weights, which means the model can pay more attention to the learning of hard samples, which benefits classification performance. In label noise training, samples with the largest uncertainty values tend to be the wrong labeled ones, so we relabel them first and pay attention to learning hard samples instead of the wrong samples.
>
> In other words, add-up loss ensures RUL can learn meaningful uncertainty values. Then the uncertainty values are used as weights to mix features to learn hard samples better and improve classification accuracy.
>
> **Sigma is set to a scalar.** We set sigma to a scalar to better represent uncertainty. As claimed in DUL [1], we use the mean value of sigma as the uncertainty value to weight the feature. A larger value ensures that the corresponding feature can take a larger part in the mixed feature; thus, the model can pay more attention to learning it in the training phase. If we weight features for each dimension, *we cannot ensure that the model pays more attention to learning the features with large uncertainty values (the mean value of sigma) as a large mean value of sigma might contain a small value in a certain dimension that is very important for classification.* Thus, using scalars to weight features is more intuitive. Actually, we have already carried out the experiment using a weighting scheme for each dimension even before writing the paper and found that the two methods perform similarly. As using a scalar to mix features can make sure meaningful uncertainty learning and conforms to the mixup framework (using sampled scalars as weights), we choose to use scalars to weight features. The experiment results are provided below, and we decide to add them to our final paper. It is shown that the per-dimension weighting scheme does not affect the performance of our method work very much. Thus, we believe that using a scalar is not limited to the extent that the features can become closer.
>
> | Method               |   acc.    | acc. with 10% rejection | acc. with 20% rejection | acc. with 30% rejection |
> | :------------------- | :-------: | :---------------------: | :---------------------: | :---------------------: |
> | per-dimension weight |   88.85   |          92.54          |          95.27          |          96.93          |
> | scalar weight        | **88.98** |        **92.72**        |        **95.40**        |        **97.35**        |
>
>
>
> **Other data domains and larger datasets.** We ran additional experiments to test the generalization ability of the uncertainty estimation to other data domains. We used pretrained models on CIFAR-10 to estimate uncertainty values for ImageNet with unseen classes in the training phase. The results are provided at https://anonymous.4open.science/r/rebuttal2021-6D55/README.md. It is shown that RUL performs better than SCN and DUL on the generalization ability of uncertainty estimation. We carried out additional experiments on AffectNet [5] and FERPlus [6]. The results show that RUL still performs well in larger datasets. We plan to include the results in our final paper.
>
> | Pretrain | Noise (%) | Method  | AffectNet |  FERPlus  |
> | :------: | :-------: | :-----: | :-------: | :-------: |
> |    x     |     0     | SCN [7] |   47.28   |   83.42   |
> |    x     |     0     |   RUL   | **49.60** | **85.21** |
> |    x     |    10     | SCN [7] |   46.29   |   78.83   |
> |    x     |    10     |   RUL   | **48.53** | **80.62** |
> |    x     |    20     | SCN [7] |   42.46   |   74.16   |
> |    x     |    20     |   RUL   | **46.04** | **78.96** |
> |    x     |    30     | SCN [7] |   40.89   |   72.71   |
> |    x     |    30     |   RUL   | **44.77** | **76.92** |
> |    √     |     0     | SCN [7] |   60.23   |   88.01   |
> |    √     |     0     |   RUL   | **61.43** | **88.75** |
> |    √     |    10     | SCN [7] |   58.60   |   84.99   |
> |    √     |    10     |   RUL   | **60.54** | **86.93** |
> |    √     |    20     | SCN [7] |   57.51   |   83.35   |
> |    √     |    20     |   RUL   | **59.01** | **85.05** |
> |    √     |    30     | SCN [7] |   54.60   |   82.20   |
> |    √     |    30     |   RUL   | **56.93** | **83.90** |
>
>
>
> We are very excited about the possibility of RUL on more tasks like modeling valence and arousal, face recognition, semantic segmentation and hope to study them in our future work.
>
> **MNIST and CIFAR.** As suggested, we ran additional experiments on ImageNet100 to better show the superiority of our uncertainty learning method. Their results have been added to our revision and summarized below.
>
> | Method  | top1 acc. (%)             0% rejection | top5 acc. (%)                   0% rejection | top1 acc. (%)    10% rejection | top1 acc. (%)   20% rejection | top1 acc. (%)   30% rejection | top1 acc. (%)  40% rejection |
> | ------- | :------------------------------------: | :------------------------------------------: | :----------------------------: | :---------------------------: | :---------------------------: | :--------------------------: |
> | SCN [7] |                 75.49                  |                    92.17                     |             74.61              |             74.49             |             74.99             |            75.37             |
> | DUL [1] |                 76.03                  |                    91.75                     |             75.80              |             79.64             |             82.86             |            85.67             |
> | RUL     |               **77.74**                |                  **93.50**                   |           **77.65**            |           **82.48**           |           **86.74**           |          **90.37**           |
>
> The results show that our method generalizes well to more difficult recognition tasks. RUL can learn useful uncertainty values to find which samples it can not correctly classify, as the accuracy rises when we reject more uncertain samples. RUL also gets the best performance without rejection showing its superiority to other state-of-the-art uncertainty learning methods.
>
> **Proof-reading.** Thanks for your sincere suggestion. We have corrected all the pointed out grammar mistakes in our manuscript and did thorough proofreading in order to improve the readability of our paper.
>
>
>
>
>
> [1] J. Chang, Z. Lan, C. Cheng, and Y. Wei. Data uncertainty learning in face recognition. In Proceedings of the IEEE/CVF Conference on Computer Vision and Pattern Recognition, pages 5710–5719, 2020.
>
> [2] Van der Maaten, Laurens, and Geoffrey Hinton. "Visualizing data using t-SNE." *Journal of machine learning research* 9, no. 11 (2008).
>
> [3] Devansh Arpit, Stanisław Jastrz˛ebski, Nicolas Ballas, David Krueger, Emmanuel Bengio, Maxinder S Kanwal, Tegan Maharaj, Asja Fischer, Aaron Courville, Yoshua Bengio, and Simon Lacoste-Julien. A closer look at memorization in deep networks. In Proceedings of the 34th International Conference on Machine Learning-Volume 70, pages 233–242. JMLR. org, 2017.
>
> [4] Chiyuan Zhang, Samy Bengio, Moritz Hardt, Benjamin Recht, and Oriol Vinyals. Understanding deep learning requires rethinking generalization. In ICLR, 2017.
>
> [5] Ali Mollahosseini, Behzad Hasani, Mohammad H Mahoor, and Mohammad H Mahoor. Affectnet: A database for facial expression, valence, and arousal computing in the wild. TAC, 10(1):18–31, 2017.
>
> [6] Emad Barsoum, Cha Zhang, Cristian Canton Ferrer, and Zhengyou Zhang. Training deep networks for facial expression recognition with crowd-sourced label distribution. In ACM ICMI, 2016.
>
> [7] K. Wang, X. Peng, J. Yang, S. Lu, and Y. Qiao. Suppressing uncertainties for large-scale facial expression recognition. In Proceedings of the IEEE/CVF Conference on Computer Vision and Pattern Recognition, pages 6897–6906, 2020.

---

> > ### Comment · Reviewer_RFnf · 2021-08-23
> > **Response to rebuttal**
> >
> > I have read the rebuttal, as well as the other reviews, and overall I feel happy with the response both to my review and to the other reviews. I believe the paper could be a good fit for the conference, despite it being targeted to a rather small community within NeurIPS audience. I acknowledge that the novelty in terms of technical details is not relevant to the very broad ML community, but I believe that it's application to the uncertainty learning is quite novel and interesting. I am not aware of any existing method that learns the classification uncertainty by mixing the features and trying to classify them simultaneously. The authors submitted the t-SNE figures I requested, and indeed Fig. 2 is quite motivating: facial expressions are in many ways subjective, and some expressions could easily have overlapping patterns. The fact that the features become far when the uncertainty is low and close when the uncertainty is high is quite revealing to me. Provided also that the method works and delivers state of the art results, I see no reasons why the paper should not be accepted. I believe that the paper has enough merits to be accepted to the conference. I will raise my rating to 7 according to these comments.

---

> > > ### Author Response · Authors · 2021-08-24
> > > **Thanks for the encouraging feedback**
> > >
> > > Thanks very much for the time and effort spent on reviewing our paper. We also thank you for your detailed reading and comments on our submission. We really appreciate your initial review, which helps us a lot to improve our work.

---

### Official Review · Reviewer_4ZJJ · 2021-07-17

**Rating:** 6
**Confidence:** 3

**Summary:**

The authors incorporated a concept of relativity into data uncertainty learning. The main method was basically a feature mixup, but the authors specially designed the mixing weights using the uncertainty values of two images.

- I feel leveraging a mixup framework for uncertainty learning is quite novel.
- The proposed method is well-motivated and sufficiently simple for people to build on.
- The proposed method works well particularly under a high level of noisy labels.
- I appreciate many visual examples.
- The paper is fairly clearly written and easy to follow.

**Ethical Concerns:**

No ethical issues

**Limitations And Societal Impact:**

The limitations of their work were not described.

**Main Review:**

[Concerns & Comments]
Though I think the main idea is quite good, the experimental evaluations concern me a lot.
- Although the proposed method looks quite general and can be applied to broad types of data, the experiments focused merely on FER. I would recommend that the authors show the wide applicability of their work by conducting experiments on face recognition datasets as in the DUL paper (Chang, CVPR'20), challenging object recognition datasets (MNIST and CIFAR10 are too easy for recent deep neural models), or others.
- The number of images in RAF-DB and FER-2013 is not large. To verify the scalability of the proposed method, some experiments on AffectNet or large-scale datasets are highly recommended.
- Relying on pretrained networks can limit the applicability of the proposed method. Could the authors train the networks from scratch and compare their results with those using pretrained networks? I think such a comparison as presented in Table 2 of the SCN paper (Wang, CVPR'20) is necessary.
- I am not sure whether the comparison in Table 4 is fair. Were the baseline models pretrained on the Ms-Celeb-1M dataset?
- It would be better if the references of baselines could be added in the cells in Tables 1, 2, and 6.

====
After the rebuttal, I raised my rating from 5 to 6 (please see the detailed comment below)

**Time Spent Reviewing:**

4

---

> ### Author Response · Authors · 2021-08-10
> **Response to Reviewer 4ZJJ's comments.**
>
> Thanks for your time and effort spent on reviewing our paper. We are sorry for omitting experiments on the datasets that you mentioned. Your thorough main review and comments are very important to the improvement of our work, which we address below:
>
> **A more challenging object recognition dataset.**  We prove the effectiveness of our method through a more challenging object recognition dataset ImageNet100. We compare our method with other uncertainty learning methods SCN [1] and DUL [2], and the results below show that our method can perform well on datasets out of the field of facial expression. The experiment results will be added to our final paper.
>
> | Method  | top1 acc. (%)             0% rejection | top5 acc. (%)                   0% rejection | top1 acc. (%)    10% rejection | top1 acc. (%)   20% rejection | top1 acc. (%)   30% rejection | top1 acc. (%)  40% rejection |
> | :-----: | :------------------------------------: | :------------------------------------------: | :----------------------------: | :---------------------------: | :---------------------------: | :--------------------------: |
> | SCN [1] |                 75.49                  |                    92.17                     |             74.61              |             74.49             |             74.99             |            75.37             |
> | DUL [2] |                 76.03                  |                    91.75                     |             75.80              |             79.64             |             82.86             |            85.67             |
> |   RUL   |               **77.74**                |                  **93.50**                   |           **77.65**            |           **82.48**           |           **86.74**           |          **90.37**           |
>
> The results show that our method generalizes well to more challenging object recognition tasks. RUL can learn useful uncertainty values to find which samples it can not correctly classify, as the accuracy rises when we reject more uncertain samples. RUL also gets the best performance without rejection showing its superiority to other state-of-the-art uncertainty learning methods.
>
> Due to time constraints, we cannot conduct experiments on face recognition datasets. We are happy to conduct experiments on face recognition datasets in the future and add the results into the supplementary material of the arXiv version.
>
> **RAF-DB and FER-2013 are not large enough.** We ran thorough experiments on AffectNet.  Results are shown in the following question and we plan to include them in the final paper.
>
> As SCN did not provide the noisy labels in AffectNet, we use the same noisy labels to train SCN and RUL to make a fair comparison. Notice that our reproduced results of SCN are slightly better than the results reported in the SCN paper. However, our method still performs better in large-scale FER datasets in all settings.
>
> **Train from scratch.** Actually, we also noticed that relying on pretrained networks can limit the applicability of our method and we have already run experiments without pertrained model on FER2013 in our paper to show the applicability of our method which is illustrated in our supplementary material Section E "FER2013 implementation details". It is claimed in line 57 of our supplementary material that " In order to verify the effectiveness of RUL when training from scratch, we carry out experiments on FER2013 without using pretrained ResNet as backbone."
>
> In order to better show the superiority of RUL, we conducted additional experiments on AffectNet and FERPlus with or without pretrained model following Table 2 of SCN as suggested. The results show that whether with or without pretrained model, RUL gets better results compared with SCN. Notice that our reproduced results of SCN are slightly better than the results reported in the SCN paper.
>
> | Pretrain | Noise (%) | Method  | AffectNet |  FERPlus  |
> | :------: | :-------: | :-----: | :-------: | :-------: |
> |    x     |     0     | SCN [1] |   47.28   |   83.42   |
> |    x     |     0     |   RUL   | **49.60** | **85.21** |
> |    x     |    10     | SCN [1] |   46.29   |   78.83   |
> |    x     |    10     |   RUL   | **48.53** | **80.62** |
> |    x     |    20     | SCN [1] |   42.46   |   74.16   |
> |    x     |    20     |   RUL   | **46.04** | **78.96** |
> |    x     |    30     | SCN [1] |   40.89   |   72.71   |
> |    x     |    30     |   RUL   | **44.77** | **76.92** |
> |    √     |     0     | SCN [1] |   60.23   |   88.01   |
> |    √     |     0     |   RUL   | **61.43** | **88.75** |
> |    √     |    10     | SCN [1] |   58.60   |   84.99   |
> |    √     |    10     |   RUL   | **60.54** | **86.93** |
> |    √     |    20     | SCN [1] |   57.51   |   83.35   |
> |    √     |    20     |   RUL   | **59.01** | **85.05** |
> |    √     |    30     | SCN [1] |   54.60   |   82.20   |
> |    √     |    30     |   RUL   | **56.93** | **83.90** |
>
> **Whether Table 4 is fair.** The comparison in Table 4 of our paper is fair. Same as the former question, in order to make a fair comparison with other methods, all experiments in our paper follow the same setting. All the results in Table 4 did not use models pretrained on the Ms-Celeb-1M dataset.
>
> **References in Tables.** Thanks for the suggestion. We have added the references in Tables 1, 2, and 6 in our manuscript as suggested.
>
>
>
> [1] K. Wang, X. Peng, J. Yang, S. Lu, and Y. Qiao. Suppressing uncertainties for large-scale facial expression recognition. In Proceedings of the IEEE/CVF Conference on Computer Vision and Pattern Recognition, pages 6897–6906, 2020.
>
> [2]J. Chang, Z. Lan, C. Cheng, and Y. Wei. Data uncertainty learning in face recognition. In Proceedings of the IEEE/CVF Conference on Computer Vision and Pattern Recognition, pages 5710–5719, 2020.

---

> > ### Comment · Reviewer_4ZJJ · 2021-08-22
> > **post-rebuttal review**
> >
> > Thank you very much for your time and effort for this rebuttal. The authors clearly resolved all the concerns regarding experiments, and I would like to raise my review score to 6 for the following reasons:
> > - The proposed RUL was further verified on AffectNet and ImageNet100 and consistently outperformed very recent baselines (DUL and SCN at CVPR'20) by a large margin.
> > - The authors showed the applicability of their method without relying on pretrained networks by conducting additional experiments on AffectNet and FERPlus.
> > - The authors also clarified that the results in Table 4 did not use pretrained models and thus the comparison was fair.
> >
> > I believe the main idea that leverages a feature mixup framework for uncertainty learning is quite novel and is well supported by experimental results. Furthermore, all the visual results including t-SNE ones are also interesting.

---

> > > ### Author Response · Authors · 2021-08-23
> > > **Thanks for the positive feedback**
> > >
> > > Dear Reviewer 4ZJJ,
> > >
> > > Thanks very much for your positive feedback and encouraging comments. Could you please kindly edit the initial review to update the rating score? If you have any further questions, we are also very glad to discuss them.
> > >
> > >
> > > Best wishes,
> > >
> > > Authors

---

> > > > ### Comment · Reviewer_4ZJJ · 2021-08-23
> > > > **RE**
> > > >
> > > > Dear AC,
> > > > I happily raised my rating from 5 to 6 in response to the authors' rebuttal. Please let me know if any approval processes and/or discussions for my action are needed.

---

### Decision · Program_Chairs · 2021-09-27

**Decision:**

Accept (Poster)

**Comment:**

After discussion, the reviewers are all for accepting the work. It's well-motivated, the idea is simple, and empirically it appears to work well in practice across a solid set of experiments. The specific novelty is fairly low, but I find this a highly overrated criteria for acceptance.  I highly recommend the authors address the rebuttal concerns as they further polish the paper.